# Effect of Various Types of Muscle Contraction with Different Running Conditions on Mouse Humerus Morphology

**DOI:** 10.3390/life11040284

**Published:** 2021-03-27

**Authors:** Kaichi Ozone, Yuichiro Oka, Yuki Minegishi, Takuma Kano, Takanori Kokubun, Kenji Murata, Naohiko Kanemura

**Affiliations:** 1Graduate Course of Health and Social Services, Graduate School of Saitama Prefectural University, Saitama 343-8540, Japan; 2191002a@spu.ac.jp (K.O.); 2191001s@spu.ac.jp (Y.O.); 2291004n@spu.ac.jp (Y.M.); 2191003x@spu.ac.jp (T.K.); 2Research Fellowship for Young Scientists, Japan Society for the Promotion of Science, Tokyo 102-0083, Japan; 3Department of Health and Social Services, Saitama Prefectural University, Saitama 343-8540, Japan; kokubun-takanori@spu.ac.jp (T.K.); murata-kenji@spu.ac.jp (K.M.)

**Keywords:** exercise, muscle contraction, bone, tendon-to-bone attachment

## Abstract

How various types of muscle contraction during exercises affect bone formation remains unclear. This study aimed to determine how exercises with different muscle contraction types affect bone morphology. In total, 20 mice were used and divided into four groups: Control, Level, Down Slow, and Down. Different types of muscle contraction were induced by changing the running angle of the treadmill. After the intervention, micro-computed tomography (Micro-CT), tartrate-resistant acid phosphatase/alkaline phosphatase (ALP) staining, and immunohistochemical staining were used to analyze the humerus head, tendon-to-bone attachment, and humerus diaphyseal region. Micro-CT found that the volume ratio of the humeral head, the volume of the tendon-to-bone attachment region, and the area of the humeral diaphyseal region increased in the Down group. However, no difference was detected in bone morphology between the Level and Down Slow groups. In addition, histology showed activation of ALP in the subarticular subchondral region in the Down Slow and Down groups and the fibrocartilage region in the tendon-to-bone attachment. Moreover, Osterix increased predominantly in the Down Slow and Down groups.Overall bone morphological changes in the humerus occur only when overuse is added to EC-dominant activity. Furthermore, different type of muscle contractile activities might promote bone formation in a site-specific manner.

## 1. Introduction

Bones play a very important role in the musculoskeletal system because they function as a lever for muscle contraction and promotion of joint movement. Exercises induce a beneficial increase in bone density during tissue maturation in growing children [1] and contribute to protection from age-related bone diseases, such as osteopenia and osteoporosis [2]. Sports activities promote bone formation, and the dominant hand humerus of a tennis player may have a larger bone circumference than the non-dominant humerus [3]. Bones cause hyperplasia when excessive mechanical stress is applied. Many in vivo animal models of various movement types, such as running, jumping, free-fall landing, and swimming, have been developed to identify the modes of movement that cause optimal bone formation [4,5,6,7]. For example, studies using treadmills found that bone formation was promoted in the group that performed flatland running [8,9].

Mechanical stress is constantly generated during exercises, and the amount of exercise is an important factor that determines the increase or decrease of mechanical stress. In the studies using the treadmill, running speed and its frequency have been often analyzed. However, it should be noted that the muscle contraction type is one of the factors that affect mechanical stress during exercises. There are roughly three types of muscle contraction, i.e., Concentric Contraction (CC), Isometric Contraction (IC), and Eccentric Contraction (EC), and CC or EC always occurs during exercise. It is known that CC has a small amount of mechanical stress because the directions of joint movement and muscle contraction are the same, while EC has a large amount of mechanical stress because the directions of joint movement and muscle contraction are opposite [10]. Therefore, when verifying the effect of mechanical stress due to exercise, it is necessary to consider not only the amount of activity but also the muscle contraction type. One clinical study found that EC-dominated training increased bone mineral density levels [11], and another study using small animals also demonstrated that an EC-dominant mode of exercise promoted femoral bone formation [12]. However, these studies did not use active exercises, and they focused only on the lower limbs in quadrupeds but not on the upper limbs, which are load joints. Moreover, very few studies have investigated the effects of changes in muscle contraction type during movement on bone-related disorders. Notably, recent evidence suggests that bone-related disorders, such as Osgood-Schlatter disease, can be caused by the misuse of the body during performance [13]. This misuse may reflect changes in muscle contraction types. Therefore, clarifying the effect of stress on bone formation based on different muscle contraction types may help elucidate the onset mechanism of sports disorders.

The purpose of this study was to examine how various types of muscle contraction associated with different running conditions affect mouse humerus morphology using microcomputer tomography (Micro-CT) and histological analysis. Because bone growth is not reflected in mature individuals, we decided to use growing mice in this study.

## 2. Materials and Methods

According to the exercise conditions that have already been established, mice were subjected to exercises that change the muscle contraction types. The following sections detail the experimental design and materials and methods.

Regarding this research, The Animal Research Ethics Committee of the authors’ university approved this study, which was performed in compliance with the University Animal Experiment Guidelines (Approval number: 2019-3).

### 2.1. Experimental Design and Exercise Intervention Protocol

Twenty male Slc:ICR mice aged 3 weeks were purchased from Japan SLC Inc. (Shizuoka, Japan). After a 1-week adaptation period, a 4-week exercise intervention was performed. All mice were individually housed in plastic cages at 23 °C ± 1 °C in a 12-h light/dark cycle. A small treadmill was used for the exercise intervention, and the mice were assigned to four groups: Control (non-running; Figure 1A) group, Level (level land running (fast speed); Figure 1B) group, Down Slow (downhill running (slow speed); Figure 1C) group, and Down (downhill running (fast speed); Figure 1D) group. Based on a previous study [14], we changed the muscle contraction type using the inclination angle of the treadmill (downhill running imitates EC of the target muscle, and level land running imitates a mixture of CC and EC of the target muscle). The shoulder joint was the target joint, and we analyzed the humeral head, supraspinatus tendon-to-bone attachment region, and humeral diaphyseal region. As EC produces approximately 1.2–2 times more joint torque and force than CC and IC, the load in the Down Slow group increases when the exercise is performed at the same speed as that in the Level group [15,16]. The running speed was adjusted for the Level and Down groups so that it was 1.5 times faster than that in the Down Slow group (Figure 1E). The exercise intervention lasted for 1 h per day and 5 days per week for 4 weeks.

### 2.2. Specimen Collection

Immediately after all exercise interventions were completed, the body mass of the mice was measured (Table 1). The animals were then euthanized by professional cervical dislocation under deep anesthesia with 2–4% isoflurane. To confirm the effect of exercise on the target muscle, the left supraspinatus muscle was peeled off from the scapula, and the wet muscle mass of the entire supraspinatus muscle was measured (Table 1). The left humerus and right shoulder joint were collected for bone morphological and histological analyses.

### 2.3. Bone Morphological Analysis

The collected left humerus was washed with physiological saline and fixed with 4% paraformaldehyde for 24 h. Micro-CT (Sky scan 1272, BRUKER, MA, USA) scan was subsequently performed (n = 5/group). The measurement condition of X-ray, detector resolution, pixel size, and rotation angle pitch were 60 kV/165 µA, 2452 × 1640, 5 µm, and 0.5°/sec, respectively. Subsequently, using a dedicated visualization application (CTvox; BRUKER, MA, USA), the measured data were converted into three-dimensional data and analyzed in each group; subsequent analysis was performed using an analysis application (CT Analyzer; BRUKER, MA, USA).

Humeral head regions were distinguished from the perfect cartilage tissue based on a certain threshold (bone: 60–200). The threshold was not changed between individuals. The cancellous bone region above the growth plate (excluding the growth plate) was used as the analysis region. The region of interest (ROI) was set using a free pen to include the entire humeral head in the sagittal section image, and the following analysis items were calculated: bone volume ratio (BV/TV), trabecular number (Tb.N), trabecular thickness (Tb.Th), and trabecular separation (Tb.Sp). In addition, for the analysis of the tendon-to-bone attachment region, which is the fibrocartilage region (including calcified fibrocartilage), the threshold was set at 60–100 to distinguish between complete cartilage and complete cortical bone. The ROI was set at the tendon-to-bone attachment region using a free pen on the sagittal section image. The range of the supraspinatus attachment (400 µm) was set as the analysis area, and its volume was calculated.

Next, for the analysis of the humeral diaphyseal region, the part 2 mm below the lower end of the surgical neck of the humerus was analyzed. Since the humeral diaphyseal region is the perfect cortical bone, the threshold was set at 100–200. The analysis items included the total cross-sectional area (Tt.Ar), cortical bone area (Ct.Ar), cortical bone area/total cross-sectional area (Ct.Ar/Tt.Ar.), and cortical bone thickness (Ct.Th). The results of Ct.Th might differ depending on the analysis part. Therefore, the thickest part, the thinnest part, and the opposite side of the thickest area in the cortical bone circumference in the analysis section were used as measurement criteria, and the average value was calculated and compared.

### 2.4. Histological Analysis

The collected right shoulder joint was fixed with 4% paraformaldehyde solution for 48 h. Subsequently, decalcification treatment was carried out with a 10% ethylenediaminetetraacetic acid solution over 2 weeks. After decalcification, the tissue was embedded in paraffin. A paraffin block was then prepared using a paraffin embedding block making device (Tissue Tech TEC™ Plus; Sakura Seiki Co., Ltd., Tokyo, Japan), and the block was stored at −20 °C until section preparation. All tissues were sliced at 5 µm thickness using a microtome REM-710 (Yamato Kohki Industrial Co., Saitama, Japan). After deparaffinization, tartrate-resistant acid phosphatase (TRAP) positive cells and alkaline phosphatase (ALP) activation were evaluated with a TRAP/ALP staining kit (FUJIFILM Wako Pure Chemical Co., Osaka, Japan). After deparaffinization, the slice was washed with distilled water. Then, a mixture of tartaric acid solution, acid phosphatase substrate solution A, and acid phosphatase substrate solution B (ratio: 10:90:1) was reacted in a 37 °C incubator for 30 min. Subsequently, the slice was washed again with distilled water and infiltrated with a 0.1 mol/L 2-Amino-2-hydroxymethyl-1,3 propanediol- hydrochloric acid solution for 10 min. It was then reacted with an alkaline phosphatase premix substrate solution for 90 min at 37 °C in an incubator. After washing again with distilled water, the slice was finally incubated with nuclear staining reagent for 5 s for counterstaining. Macroscopic observations were performed to analyze the histological images in the articular cartilage region and the tendon-to-bone attachment region.

In addition, immunohistochemical (IHC) staining was performed for the humeral head to label osteoblast differentiation marker Osterix (OSX) to evaluate the dynamics of osteoblasts. First, tissue sections were washed three times for 5 min with a PBS (pH, 7.4) solution. For the antigen activation treatment, a Proteinase K (Worthington Biochemical Co., NJ, USA)/distilled water solution (0.2 mg/mL) was added dropwise onto the sections and incubated for 15 min. After washing with PBS (three times for 5 min), endogenous peroxidase was inactivated by incubating the sections in 0.3% hydrogen peroxide/methanol solution for 30 min. After washing again with PBS (three times for 5 min), sections were blocked with 5% normal goat serum/PBS solution and were then incubated with anti-Osterix (OSX) rabbit polyclonal antibody (1:300 dilution, bs-1110R, Bioss, MA, USA) overnight at 4 °C. The streptavidin-biotin-peroxidase complex technique was then performed at room temperature using an ABC kit (Vector Laboratories, CA, USA). After washing with PBS (three times for 5 min), sections for immunohistochemical analysis were stained using diaminobenzidine (Agilent Technologies, CA, USA), and counterstaining of the nuclei was performed with 25% Mayer’s hematoxylin. Image analysis software Fiji [17] was used to analyze IHC-stained images. The analysis areas were the tendon-to-bone attachment subchondral bone region and the subchondral bone region of the articular surface, where a load is easily applied, and the area of positive cells per unit area was calculated.

### 2.5. Statistical Analysis

All analyses were performed using R version 3.4.3 (The R Foundation for Statistical Computing, Vienna, Austria). The Shapiro–Wilk test was used to determine the normality of distribution for each dataset, and normality was recognized in all analyses. One-way analysis of variance was performed after conducting the normality test. Tukey’s honestly significant difference method was used for posthoc analysis. A *p*-value of < 0.05 was considered significant. All results are shown as mean ± standard deviation (SD).

## 3. Results

The comparison results of different groups that received exercise intervention are shown below. Bone morphological results were classified by the humerus head region and the humerus diaphyseal region. The histological analysis result is shown only for the humeral head region.

### 3.1. Body and Muscle Mass Comparison Results

The results of comparisons of body mass after all interventions, wet mass of the supraspinatus muscle, and wet mass of the muscle corrected by the body mass are shown in Table 1. Regarding body mass, a significant difference was detected only between the Control and Down groups (*p* < 0.05, Table 1). In addition, the supraspinatus muscle wet mass was significantly different between the Control and Down Slow groups, between the Control and Down groups, between the Level and Down Slow groups, and between the Level and Down groups (*p* < 0.05, Table 1). After normalizing the wet muscle mass by body mass and calculating the relative value, significant differences were still confirmed between the Control and Down Slow groups, between the Control and Down groups, between the Level and Down Slow groups, and between the Level and Down groups (*p* < 0.05, Table 1).

### 3.2. Bone Morphological Results

#### 3.2.1. Humeral Head and Tendon-to-Bone Attachment Regions

The humeral head region imaged by Micro-CT is shown in Figure 2A. Significant differences in BV/TV and Tb.Th were detected only between the Control and Down groups (*p* < 0.05, Figure 2B,C). No significant difference was found in Tb.N between all groups (*p* = 0.09, Figure 2D). Moreover, significant differences in Tb.Sp were found between the Control and Down groups (*p* < 0.05, Figure 2E) and between the Level and Down groups (*p* < 0.05, Figure 2E). The most significant difference in the FC volume of the supraspinatus tendon-to-bone attachment region was detected between the Control and Down groups (*p* < 0.01, Figure 2F), followed by differences between the Control and Level groups (*p* < 0.05, Figure 2F) and between the Level and Down groups (*p* < 0.05, Figure 2F). Macroscopic observation showed that the subchondral bone region on the humeral joint surface (white arrowhead) appeared to increase in all intervention groups compared to the Control group. There was also a difference in the subchondral bone region among the exercise groups; the Down Slow group showed an increasing tendency than the Level group, and the Down group showed an increasing tendency than the Down Slow group.

#### 3.2.2. Humeral Diaphyseal Region

The humeral diaphyseal region imaged by Micro-CT is shown in Figure 3A. Tt.Ar was not significantly different among all groups (*p* = 0.53, Figure 3B). However, significant differences in Ct.Ar were found between the Control and Down groups (*p* < 0.01, Figure 3C) and between the Level and Down groups (*p* < 0.05, Figure 3C). Therefore, a significant difference in Ct.Ar/Tt.Ar was observed only between the Control and Down groups (*p* < 0.01, Figure 3D). Furthermore, Ct.Th significantly differed between the Control and Down groups (*p* < 0.01, Figure 3E) and between the Level and Down groups (*p* < 0.05, Figure 3E).

### 3.3. Histological Results in the Humeral Head Region

TRAP/ALP staining was performed for the humeral head, and macroscopic observation was conducted. In the articular cartilage region, TRAP-positive cells showed no changes in all groups. ALP active cells were almost not detected in the Control group but were observed in the articular cartilage region in the Level group. In the Down Slow and Down groups, ALP active cells were found not only in the articular cartilage region but also in the subchondral region, and the osteogenic reaction was also detected. Enlargement of the subchondral bone region was found as the osteogenic reaction increased (Figure 4A). In addition, similar to the observation results in the articular cartilage, in the tendon-to-bone attachment region, TRAP-positive cells showed no changes in all groups. ALP-active cells were almost not found in the subchondral bone region but were detected in the calcified fibrocartilage region of the tendon-to-bone attachment region. Although no changes in ALP activation were observed between the Contol and Level groups, ALP was activated more in the Down Slow and Down groups than in the other groups (Figure 4B).

OSX-labeled IHC staining was performed for the humerus head (Figure 5). The OSX-positive areas in the articular cartilage subchondral bone region were 0.92% ± 0.47 in the Control group, 1.10% ± 0.49 in the Level group, 2.15% ± 0.54 in the Down Slow group, and 2.48% ± 0.69 in the Down group (Figure 5A). Significant differences were observed between the Control and Down Slow groups (p < 0.001; Figure 5C), the Control and Down groups (*p* < 0.001; Figure 5C), the Level and Down Slow groups (*p* < 0.01; Figure 5C), and the Level and Down groups (*p* < 0.001; Figure 5C). The OSX-positive areas in the tendon-to-bone attachment subchondral bone region were 1.17% ± 0.30 in the Control group, 1.30% ± 0.35 in the Level group, 1.93% ± 0.33 in the Down Slow group, and 2.15% ± 0.28 in the Down group (Figure 5B). Significant differences were found between the Control and Down Slow groups (*p* < 0.001; Figure 5D), the Control and Down groups (*p* < 0.001; Figure 5D), the Level and Down Slow groups (*p* < 0.01; Figure 5D), and the Level and Down groups (*p* < 0.001; Figure 5D).

## 4. Discussion

This study clarified the effects of different exercise-associated muscle contraction types on the humeral head and humeral diaphyseal of mice by bone morphological and histological analyses. Our hypothesis was that the effect of exercises on bone during growth depends on the type of muscle contraction during movement, and that significant changes occur regardless of the amount of exercises, especially in the EC-dominated group. However, the bone morphology analysis found that only the volume of the tendon-to-bone attachment region and Tb.Sp in the humeral head showed significant changes in the Down Slow and Down groups, and the Down Slow group did not show any specific changes in other items. In addition, histological analysis demonstrated site-specific changes in the loaded articular cartilage subchondral bone region, and OSX, a marker of osteoblast differentiation, was confirmed at similar sites. Interestingly, there were no major changes between the control and level groups.

In this study, total travel distances during the intervention period were about 27 km in the Level and Down groups and about 18 km in the Down Slow group. The average running speeds were 22.5 m/min in the Level and Down groups and 15 m/min in the Down Slow group. Although the average intensity was moderate, the speed gradually increased with age; the exercise intensity was limited to the extent so that no mice dropped out after one hour of running (maximum 30 m/min at eight weeks of age). In addition, although the running speed differed between the Level and Down Slow groups, the exercise intensity was almost the same considering the effect of torque [15,16]. Therefore, if there are significant differences between these groups, the muscle contraction type might have an effect.

In the overall humeral head, the number of trabecular bones in the subchondral bone region was not affected in the Down group compared with the other groups, and the expansion of trabecular thickness and the narrowing of the trabecular space were promoted in the Down group, thereby leading to an increased bone volume ratio. A characteristic change was observed in the subchondral bone region just below the articular surface, which is predicted to be loaded, and this region was thickened in the Down Slow and Down groups compared with the Level group. Another characteristic change was found in the tendon-to-bone attachment region, which significantly increased in the Down Slow and Down groups compared with the Control group and significantly differed between the Down and Level groups. The same tendency was obtained from histological observations; in the Down Slow and Down groups, an increase in ALP active cartilage cells and a positive osteogenic reaction in the subchondral bone region (localized to the articular cartilage region under load) were confirmed, and the number of OSX-positive cells, which are markers of differentiation into osteoblasts, increased. The width of the subchondral bone also increased in the Down Slow and Down groups compared to the Control and Level groups. ALP activity is positive not only in cartilage calcification but also in mature osteoblasts [18], and OSX is positive when preosteoblasts differentiate into mature osteoblasts [19]. These findings may indicate that osteoblasts increased in the subchondral bone region, and bone formation was promoted. In addition, ALP activation tended to increase in the fibrocartilage area of the tendon-to-bone attachment region in the Down Slow and the Down groups, indicating that the calcification reaction in the fibrocartilage area is promoted; this finding is similar to the volume result of the tendon-to-bone attachment region in the bone morphological analysis. Promotion of calcification reaction in the fibrocartilage area may have been caused by an increase in OSX in the subchondral bone region. In fact, past studies have also confirmed overexpression of OSX in the subchondral bone region of tendon-to-bone attachment when the tendon-to-bone attachment region causes pathological ossification [20]. Furthermore, in the diaphyseal region, Ct.Ar/Tt.Ar showed a significant increase in the Down group compared to the Control group. In detail, although no significant difference was confirmed in Tt.Ar in all groups, Ct.Ar and Ct.Th were significantly increased in the Down group compared with the Control and Level groups. In other words, the cortical bone area and width increased in the total cross-sectional area of the bone in the Down group. Despite no significant difference, the change in the Down Slow group showed the same tendency as that in the Down group.

The forelimbs of quadrupeds are load-bearing joints; when traveling downhill, grand reaction force (GRF) of the forelimbs increases, and the forelimbs account for approximately 84% of the total braking impulse during downhill running [21]. When the amount of mechanical stress applied to bone increases, a pressure electric potential is generated on the bone surface, and cytokines involved in bone remodeling are secreted from a complex composed of osteoblasts and bone cells. At the same time, increased blood flow in the bone directly and jointly promotes osteoblast activation and bone formation [22,23]. Although the blood flow in the bone did not be determined in this study, it was confirmed that changes in osteoblasts were significantly increased in the Down Slow and Down groups. Thus, it is highly possible that the same phenomenon occurred. Moreover, it has also been demonstrated that simply increasing the amount of exercise load changes the thickness of the subchondral bone. For example, Murray et al. showed that high-intensity exercise in horses resulted in increased subchondral bone thickness, enhanced bone modeling, and decreased bone resorption at high-load sites [24]. In addition, the same result was found in an experiment that imitated high-intensity exercise by running downhill [25]. It is known that the cortical bone in the normal bone stem becomes thicker with growth, and the thickness of the cortical bone adapts to the stress and increases when stress in the long axis direction is applied. Therefore, bone formation is promoted after exercises, such as jumping exercises, in which pressure stimulation increases [26,27]. Based on our present findings, the Down group, which had high-intensity exercise conditions, showed a significant change in bone morphology compared to the other groups. Therefore, it is possible that bone morphology in the Down group changed significantly due to increases in longitudinal mechanical stress and accompanied GRF and excessive exercise conditions that promoted the activation of osteoblasts and osteocytes.

This study set such conditions that EC, which shows a muscle hypertrophy effect by carrying out downhill running, became dominant [28]. The wet mass of the muscle corrected by the body mass increased significantly in the Down Slow and Down groups compared to the other groups, indicating that the muscles have become hypertrophied. It is known that exercise in EC causes muscle hypertrophy, and the tension exerted to the bone increases as the muscles become hypertrophied [29]. Since there was a significant difference between the Level and Down slow groups in which the exercise intensity was adjusted, it is possible that the tension applied to the bone was higher in the Down slow group than in the Level group even with low-load EC exercise. In fact, many studies have shown that muscle volume affects bone formation, and reducing mechanical stress through hind limb suspension and paralysis promotes muscle volume and bone atrophy. Conversely, bone hyperplasia occurs by increasing muscle volume using a model lacking myostatin [30,31,32]. Similarly, the mechanical stress associated with muscle contraction is essential for formation of tendon-bone attachment regions and promotion of mineralization, and they are not normally formed when the mechanical stress is attenuated in conditions, such as paralysis [33,34]. Taken together, increased mechanical stress due to different running conditions and contraction types can lead to hypertrophy of the target muscle, thereby promoting humerus bone formation and increased volume of tendon-to-bone attachment regions.

Unfortunately, the results of this study did not completely support our initial hypothesis. No bone morphological changes were observed between the Level and Down Slow groups, although they showed muscle hypertrophy. Therefore, we can not conclude from this study that changes in muscle contraction type affect bone morphology regardless of the amount of exercise. However, when observed by site, the subchondral bone region tended to expand in the region where the load was likely to be applied, and it was confirmed that the ALP activity and expression of Osterix (a marker of osteoblast differentiation) increased. Similar results were obtained in the tendon-bone attachment region, suggesting that molecular fluctuations might occur in a site-specific manner, although there is no morphological change.

In this study, we confirmed changes in the humerus due to changes in the muscle contraction type. This study suggests that the tendon-to-bone attachment region may change in a muscle contraction type-specific manner; this is a very valuable clinical result. Osgood-Schlatter and Sever diseases are disorders of the tendon-to-bone attachment region and have been recognized to be caused by muscle overuse, especially in young sports activists [35]. In recent years, clinical studies have demonstrated that misuse of the body during movement may be involved in the onset of Osgood-Schlatter and Sever diseases, but basic research has not reported any related evidence. In this study, bone morphological changes occurred, in addition to ALP activity and OSX expression, which promote bone formation, in the group (Down) that was overused and mimicked EC. However, the volume of the fibrocartilage region increased slightly, and the OSX expression and ALP activity were significantly increased even in the group with low amount exercise and imitating EC (Down Slow) compared to the other groups. Therefore, the morphological change of the tendon-to-bone attachment region may be due to the change of the muscle contraction type rather than overuse. In the future, focusing on changes in the tendon-to-bone attachment region and clarifying the relationship between muscle contraction type and bone formation in more detail may help elucidate the pathogenic mechanism of sports disorders.

This study has some limitations. First, this study focused on bone morphological analysis and histological analysis, and we did not determine whether increased mechanical stress leads to molecular and gene changes. In addition, although the muscle mass was measured, the amount of mechanical stress actually applied was unknown because the muscle tension test could not be performed. Furthermore, we could not determine whether the bone morphological changes are functionally strong. Therefore, future studies are needed to further clarify the relationship between muscle contraction types and bone formation by muscle tension test, molecular biological analysis, and dynamic bone formation measurement. To our knowledge, very few studies have determined the association between muscle contraction types and bone formation during exercise. Particularly, changes in the tendon-to-bone attachment region have not been analyzed to date. This study provides additional data on the mechanisms underlying exercise-associated bone changes.

## 5. Conclusions

This study found that when EC was induced by running downhill, muscle mass tended to increase. In addition, we observed bone morphology changes in the head of humerus, tendon-to-bone attachment region, and diaphyseal region of the humerus and an increase in osteoblasts. However, when a mixture of EC and CC was induced in the Level group, no change was shown. Furthermore, EC-dominated slow movement (Down Slow) tended to increase muscle mass and osteoblasts but did not change bone morphology. Our findings demonstrate that overall bone morphological changes in the humerus occur only when overuse is added to EC-dominant activity, and EC activity might promote bone formation in a site-specific manner, even at a low load.

## Figures and Tables

**Figure 1 life-11-00284-f001:**
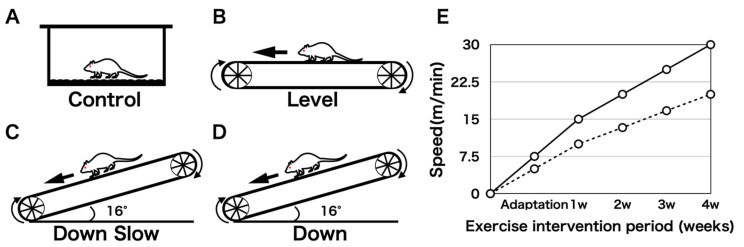
Grouping and exercise intervention protocol. (**A**) Control group: free movement in the cage, non-running intervention. (**B**) Level group: running on the level land and fast speed running intervention. (**C**) Down Slow group: running downhill and slow running speed intervention. The predominance of eccentric contraction (EC) during running exercise increases. (**D**) Down group: running downhill with the same intervention speed as that in the Level group. The predominance of EC during running exercise increases. (**E**) Temporal changes in exercise intervention speed. The solid line shows changes in exercise velocity in the Level and Down groups. The dotted line shows changes in exercise velocity in the Down Slow group. Considering exercise intensity, the Level and Down groups were subjected to interventions at a rate 1.5 times faster than that in the Down Slow group.

**Figure 2 life-11-00284-f002:**
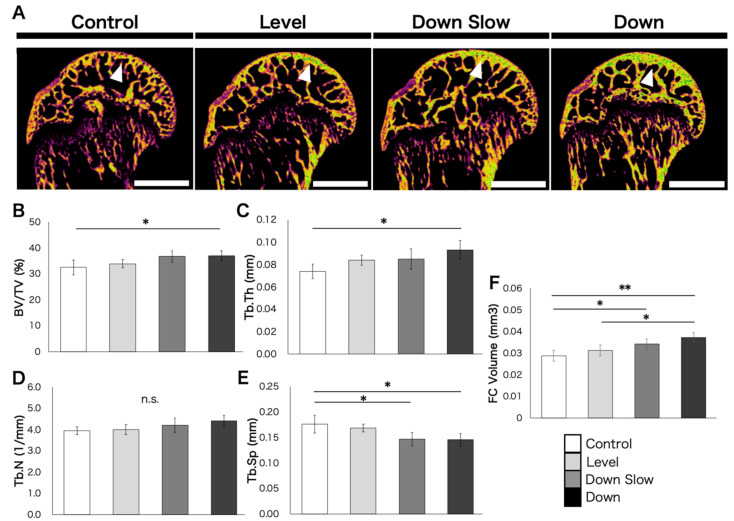
Bone morphological changes in the humeral head region. (**A**) Micro-CT image of the humeral head in each group. Scale bar, 1 mm. White arrowhead, the subchondral bone region on the humeral head joint surface. (**B**) Comparison results of the bone volume ratio (BV/TV) in the head of the humerus. (**C**) Comparison results of the trabecular thickness (Tb.Th) in the head of the humerus. (**D**) Comparison results of trabecular number (Tb.N) in the head of the humerus. (**E**) Comparison results of trabecular separation (Tb.Sp) in the head of the humerus. (**F**) Morphology comparison results in the supraspinatus tendon-to-bone attachment fibrocartilage (FC) region. (**B**–**F**) Data are shown as mean ± SD. *, *p* < 0.05. **, *p* < 0.01.

**Figure 3 life-11-00284-f003:**
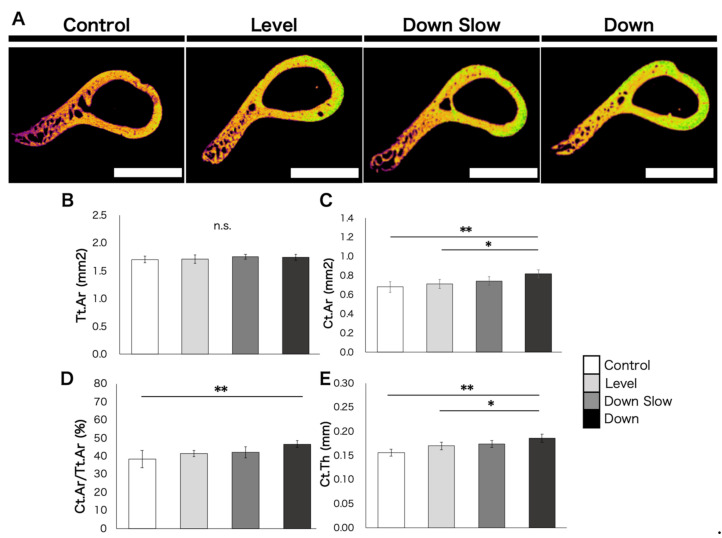
Bone morphological changes in the humeral diaphyseal region. (**A**) Micro-CT image of the humeral diaphyseal in each group. Scale bar, 1 mm. (**B**) Comparison results of the total cross-sectional area (Tt.Ar) in the humeral diaphyseal. (**C**) Comparison results of the cortical bone area (Ct.Ar) in the humeral diaphyseal. (**D**) Comparison results of the cortical bone area/total cross-sectional area (Ct.Ar/Tt.Ar) in the humeral diaphyseal. (**E**) Comparison results of cortical bone thickness (Ct.Th) in the humeral diaphyseal. (**B**–**E**) Data are shown as mean ± SD. *, *p* < 005. **, *p* < 0.01. n.s., *p* > 0.05.

**Figure 4 life-11-00284-f004:**
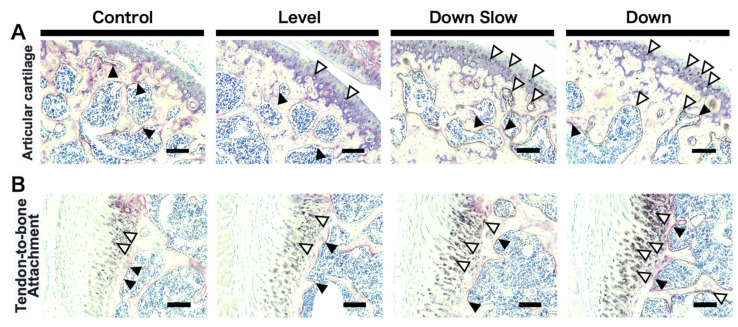
Tartrate-resistant acid phosphatase (TRAP)/alkaline phosphatase (ALP) double staining and macroscopic observations in the humeral head region. (**A**) TRAP/ALP double stained images in the articular cartilage region. (**B**) TRAP/ALP double stained images in the tendon-to-bone attachment region. (**A**,**B**) Black arrowhead: TRAP-positive cells. White arrowhead: ALP active cells. Scale bar: 100 µm.

**Figure 5 life-11-00284-f005:**
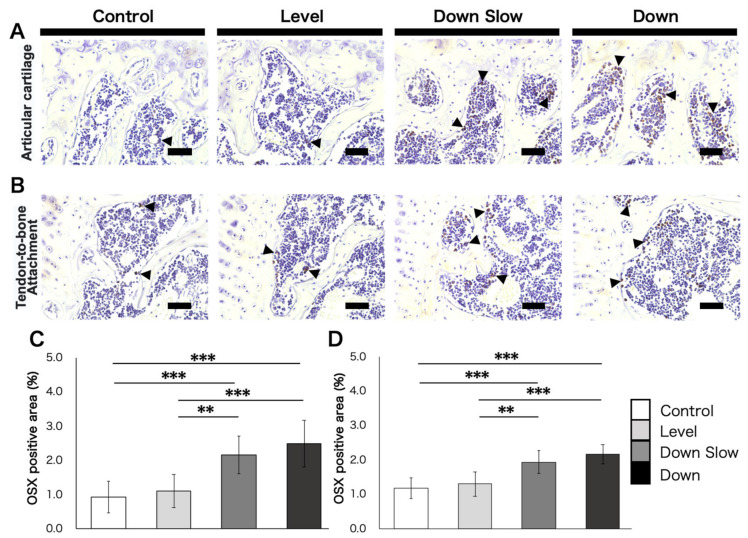
Immunohistochemical (IHC) staining of the humeral head region (OSX labeling). (**A**) IHC-stained images in the articular cartilage subchondral bone region. (**B**) IHC-stained images in the tendon-to-bone attachment subchondral bone region. (**A**,**B**) Black arrowhead: OSX-positive cells. Scale bar: 50 µm. (**C**) Comparisons of the OSX positive area in the articular cartilage subchondral bone region. (**D**) Comparisons of the OSX positive area in the tendon-to-bone attachment subchondral bone region. (**C**,**D**) Data are shown as mean ± SD. **, *p* < 0.01; ***, *p* < 0.001.

**Table 1 life-11-00284-t001:** Overall Body Mass and Wet Mass of the Supraspinatus Muscle.

Group	Body Mass (g)	Muscle Wet Mass (mg)	Normalized Value (mg/g)
Control	35.4 ± 1.02	46.0 ± 4.90	1.30 ± 0.12
Level	35.6 ± 1.02	50.0 ± 6.32	1.40 ± 0.17
Down Slow	36.8 ± 0.75	62.0 ± 4.00 *^,^^‡^	1.68 ± 0.08 *^,^^‡^
Down	37.4 ± 1.02 ^†^	64.0 ± 4.90 ^†,§^	1.70 ± 0.09 ^†,§^

Data are shown as mean ± SD. * Control vs. Down Slow group, *p* < 0.05. ^†^ Control vs. Down group, *p* < 0.05. ^‡^ Level vs. Down Slow group, *p* < 0.05. ^§^ Level vs. Down group; *p* < 0.05.

## Data Availability

Not applicable.

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
