# Peer review of "Effect of Various Types of Muscle Contraction with Different Running Conditions on Mouse Humerus Morphology"

_life, 2021, doi:10.3390/life11040284_

Round 1

Reviewer 1 Report

The MS is questioning how various types of muscle contraction affect bone formation. This is an interesting and important physiological point but the answer is still not clear. The authors present their methodology of mice training to investigate how the variation in the running conditions (no running, Control group, on a flat surface, Level group,  or downhill at different speeds, Slow Down and Down groups) affect growth of humerus and shoulder joint. Post-mortem morphological and histological analyses are presented.

 The text is clearly written and the data is well described. To the authors and my surprise no significant difference in any of measured characteristic was found between the Level and Down Slow groups, which may mean that the induced stress in slow downhill running was not enough. While between Down Slow and Down groups the difference in bone structure was statistically significant.  The experimental work lacks biomechanical measurements but the authors are aware of it.

Anyhow the detailed and fair discussion of all observed changes is given and I hope these studies will be combined with the studies of muscle mass and force and continued to clarify the role of amount and type of muscle contraction in bone formation.

Reviewer 2 Report

The publication presents the important topic of the effect of various types of muscle contraction with different running conditions on the humerus morphology in mice.

The topic is very interesting and worth research, but before publishing the following manuscript, I recommend minor corrections:

In the introduction, the importance of the undertaken research was insufficiently emphasized. This should be developed.

The Materials and Methods section does not provide a legal basis for animal testing, ethics committee approval, etc.

The methodology, in particular the histology, is very poorly described. Please specify the specific steps for securing the tissue in paraffin, staining etc.

There are two figures 4, no figure 3. I assume that this is a mistake. Both of them, the figure 3 and 4 should be improved. The graphs are too small and it is hard to read anything from them.

The results are very poorly discussed with similar papers. In the second part of the discussion, the authors refer to the papers analyzing molecular factors, changes of which were accompanied by bone remodeling. However, they themselves do not investigate such factors, which confuses the reader. Undoubtedly, undertaking research at the molecular level, examining, for example, angiogenesis in muscles or analyzing cytokines present during exercise, e.g. from blood, would be interesting and would increase the level of research.

Reviewer 3 Report

The manuscript “Effect of various types of muscle contraction with different 2 running conditions on mouse humerus morphology” by Ozone et al. describes the effect of differential muscle contractions during exercises on bone morphology. Overall, the manuscript is well-written and relevant questions are asked. However, only 5 mice are used per mouse cohort and that number is quite low.

The major weakness of the paper is that the study lacks convincing biochemical or molecular evidence. Authors have only used micro-computed tomography (micro-CT) and histological analysis to read their data. Furthermore, if authors could supplement the manuscript with some data to back their claims, the quality of the manuscript will be enhanced.

Most importantly, clinical significance of the disease and why the study was done in should be explained very clearly in the.  

Round 2

Reviewer 3 Report

Authors have added additional information and have satisfactorily answered the queries.